

# Relationship between impulsivity and suicide among the rural elderly in China: a case-control psychological autopsy study

Yunfang Zhou[1]   Zhenyu Ma[2]   Cun-Xian Jia[3]   Liang Zhou[4]

[1] Department of Labor and Social Security, School of Public Administration, Hunan University of Finance and Economics, Changsha, China

[2] School of Public Health, Guangxi Medical University, Nanning, China

[3] School of Public Health, Shandong University, Jinan, China

[4] The Affiliated Brain Hospital of Guangzhou Medical University (Guangzhou Huiai Hospital), Guangzhou, China

## ABSTRACT

**Background.** The relationship between impulsivity and suicide is inconsistent in different populations. Hence, the relationship between impulsivity and suicide still needs to be studied among the elderly population. The present study intends to explore the relationship between impulsivity and suicide among the rural Chinese elderly.

**Methods.** A case-control psychological autopsy study was conducted from February 1, 2014 to December 18, 2015 among rural residents over the age of 60 who died by suicide. The sample consisted of 242 suicides as the case group and 242 living individuals as the control group. Data on demographic characteristics, impulsivity, previous history of suicide attempts, social support, negative life events, and suicidal behavior were collected.

**Results.** Our study found that impulsivity increased the risk of suicide. The case group showed a higher Barratt Impulsiveness Scale score compared with the control group ($p < 0.001$), which indicates that impulsivity was higher among the elderly suicides. In addition, regression analyses show that impulsivity (odds ratio: 1.03, 95% confidence interval: 1.01–1.06) is an independent risk factor of suicide, after controlling for the effects of marital status, education, family annual income, being left behind, social support, and negative life events. Finally, compared with elderly who do not have a history of attempted suicide, elderly with a history of attempted suicide showed higher impulsivity ($p = 0.001$).

## INTRODUCTION

Suicide, which is the act of intentionally taking one's own life (*Turecki et al., 2019*), has been considered as a major global health problem (*Dong et al., 2015*). Suicide causes 800, 000 deaths every year and has been listed as the 15th leading cause of death all over the world (*WHO, 2019*). Although the global prevalence of suicide has remarkably dropped in recent years, suicide in China contributes to the overall burden of suicide, accounting

Corresponding author
Liang Zhou,
Liangzhou_csu@vip.163.com

for approximately 1/5 of global suicides (*Dong et al., 2015*). In China, the risk of suicide increases with age, with elderly aged over 65 having the highest suicide rate of 44.3-200 per 100,000, which is 4-5 times higher than the general population (*Li, Xiao & Xiao, 2009*). Despite the decreasing suicide rate among the general population, suicide among the elderly population has shown an increasing trend (*Sha, Yip & Law, 2017*; *Wang, Chan & Yip, 2014*). Moreover, the suicide rates of elderly in rural areas are higher than those in urban areas in China (*Li & Katikireddi, 2019*). China has a rapidly ageing population, but the mental health care system in rural China is still in its infancy. In addition, most rural areas of China are experiencing high work migration where young people move to big cities to look for job opportunities and leave their old parents living in countryside alone. Rural elderly, especially those living alone, are faced with tremendous difficulties and challenges physically, mentally and socially, which put them at higher risk of suicide. Consequently, it is both important and urgent to understand contributing factors that lead to suicide, which is rarely studied among rural elderly population in China.

Among a range of risk factors that have been identified to predict suicide, impulsivity is one of the most-commonly mentioned, yet also most controversial factor reported in the literature. There is no standard definition of impulsivity, which has been defined and operationalized in various ways by various researchers (*Bakhshani, 2014*). *Moeller et al. (2001)* looked at impulsivity from a bio-psycho-social perspective and emphasized three essential aspects of impulsivity: (1) decreased sensitivity to negative consequences of behavior; (2) immediate and unplanned reaction to stimuli before processing the information thoroughly; and (3) no regard for long-term consequences of a behavior. This definition provides helpful insight in guiding for research and treatment on impulsivity and its related risk behaviors.

Although impulsivity has been widely recognized as a predisposition towards risky behaviors such as suicide (*Bakhshani, 2014*), studies on the relationship between impulsivity and suicide have yielded inconsistent and variable results. A large number of research have adopted impulsivity as a significant risk factor or waning sigh for suicide. For instance, a literature review by *Gvion & Apter (2011)* showed that impulsivity was highly correlated with suicidal behavior across both psychiatric and non-psychiatric populations. Another literature review and meta-analysis on impulsivity in the self-harm and suicidal behavior of young people also demonstrated significant positive associations between multiple facets of impulsivity with suicide behaviors (*McHugh et al., 2019*). As a result, impulsivity has been highlighted as an important suicide risk factor by many organizations including the American Association of Suicidology, American Foundation for Suicide Prevention and the Substance Abuse and Mental Health Services Administration. However, an almost equally large number of studies have also shown small or even no association between impulsivity and suicide. For example, *Smith et al. (2008)* posited that most suicide is not a result of impulsive decisions but involve a plan. This proposition was also endorsed by *Anestis et al. (2014)* who conducted a literature and meta-analysis of the association between trait impulsivity and suicidal behavior and found very small correlations. A recent literature review also found only limited between-group differences in various aspects of impulsivity between suicide and non-suicide group (*Beach, Gissandaner & Schmidt, 2021*).

Furthermore, *Klonsky & May (2015)* found no association between impulsivity and suicide in both their own studies and by literature review.

The inconsistent results on impulsivity and suicide may be partly explained by differences in sample, suicide outcomes (ideation, attempt, and death), measurement of impulsivity, and study design across various studies. However, there have been no studies that examined the association between impulsivity and suicide among older adults in China, who are most susceptible and vulnerable to suicide. A better understanding of suicide and its association with impulsivity among Chinese rural elderly not only contributes more evidence to the already conflicting literature, but also help guide for further intervention programs for the treatment of impulsivity and prevention of suicide. The current study was conducted to fill in the research void by investigating the relationship between suicide and impulsivity among Chinese rural older adults.

## METHODS

### Design and ethical statement

The study, conducted from February 1, 2014 to December 18, 2015, utilized a matched case–control study design combined with psychological autopsy method. The study was approved by the Institutional Review Board of the Central South University (Ethical Application Ref: CTXY-130041-1), Shandong University (Ethical Application Ref: 20150306-1), and Guangxi Medical University (Ethical Application Ref: 20150146).

### Sample and sampling

Data on rural residents over the age of 60 who died by suicide were collected in this study. Information on suicide deaths among the elderly, which was used as the case group, was obtained from the local governments' death certification system. For the control group, living comparisons were matched 1:1 with the suicides in the case group in terms of age (±3 years), gender, and residence. When a suicide from the case group is identified, we first listed and enumerated all the elderly adults from the same village with matching age and gender. Then, a computer software was used to randomly select one living comparison from the list of matching individuals. In rare instances when no appropriate living comparisons were available, we expanded our search to the nearest villages.

A multistage stratified cluster sampling method was adopted to recruit participants. First, Shandong, Hunan, and Guangxi provinces were selected to correspond to eastern, central, and western China, respectively. Second, 12 counties were randomly chosen from the three provinces, including Zoucheng, Junan, and Gaotang from Shandong Province; Yongding, Cili, and Sangzhi from Hunan Province; and Hengxian, Wuming, Lingchuan, Hepu, Luzhai, and Lingyun from Guangxi Province. Elderly residents were recruited, and information on elderly suicides was obtained from each county. Finally, 242 suicides and 242 living individuals were included in this study.

### Procedure of the interview

Face-to-face interviews were conducted with two informants of both suicide group and living controls. The two informants usually included one next-to-kin who lived with the

subjects, and one non-kin person such as friend, neighbor, or a remote relative. Interviewers from the three research sites received uniform and standard training on psychological autopsy and interview methods before the start of the interviews. Each interview took about 90 min with one interviewer interviewing one informant each time. The aims and procedures of the research were explained, and a written informed consent was obtained from the participants at the start of the interview. Each interview lasted approximately 90 minutes. The details of the participant selection and interview procedures were described in previous publications (*Zhou et al., 2019*).

## Measurements
### Demographic characteristics
The demographic data included gender, age, educational level, marital status, family income, physical diseases, and being left behind. Marital status was classified into currently married (including married, remarried and cohabiting) and not currently married (including single, separated, divorced, and widowed). Being left behind is defined as desertion by adult children 12 months prior to suicide (for the case group) or investigation (for the control group), residence away from the original township for at least 10 months, and infrequent visits (no more than twice).

### Barratt impulsiveness scale
Impulsivity was measured using the Barratt Impulsiveness Scale (BIS). It is a 30-item scale designed to assess the personality and behavioral construct of impulsiveness and includes three subscales: non-planning, motor, and attentional impulsivity (*Patton, Stanford & Barratt, 1995*). Each item is rated on a five-point Likert scoring from 1 "never" to 5 "very frequently". The total score ranges from 30 to 150, with a higher score indicating high impulsivity. The original BIS showed good internal consistency with Cronbach's alpha of 0.77-0.89 and a test–retest reliability of 0.68-0.89 (*Patton, Stanford & Barratt, 1995*). The Chinese version of BIS has been used in a previous psychological autopsy study in China and showed satisfactory reliability and validity (*Lu et al., 2012*).

### Life events scale for the elderly (LESE)
Life events were measured using the Life Events Scale for the Elderly (LESES). It is 46-item scale specifically developed for elderly adults to assess stressful life events during the last 12 months before suicide or investigation and includes three subscales: health-related events, family problems, and social communication problems. Each event was recorded from five dimensions: the date it happened, whether it was positive or negative, the impact to mental health of the target person, the duration of the event, and the number of times it happened. The intensity of each life event is calculated by the impact multiplied by the duration and then by the number of times it happened. The total score of negative life events is the sum of all the scores of negative life events, with higher score indicating more negative life events. The Chinese version of LESE has been used in a previous psychological autopsy study in China and showed satisfactory reliability and validity (*Mo et al., 2019*).

### Duke social support index

Social support was measured using the Duke Social Support Index (DSSI). It is a 23-item scale designed to assess multiple dimensions of social support in the last week before death/investigation. The total score ranges from 11 to 45, with a higher score indicating higher social support. The Chinese version of DSSI has been used in a previous psychological autopsy study in China and showed satisfactory reliability and validity (*Pan et al., 2020*).

### Suicide intent scale

The Suicide Intent Scale (SIS) was used to measure factual aspects of the suicide sttempt, which included the attempters' precautions, planning, communication, and expectations about the suicide behavior (*Beck & Lester, 1976*). The Chinese version of SIS has been used in a previous psychological autopsy study in China and showed satisfactory reliability and validity (*Ma et al., 2020*; *Zhang & Jia, 2007*).

### Features of suicide behavior and history of suicide attempts

Information on the time, place, and method of suicide in the case group were collected. The data obtained included the number of suicide attempts and the timeframe and method of the most recent suicide attempt.

### Statistical analysis

The data were analyzed by using SPSS 19.0 software package. Comparisons of parametric data for the two groups were performed using one-way blocked analysis of variance and chi-squared test. The Wilcoxon signed-rank test was used for nonparametric data in the two groups.

Conditioned multivariable logistic regression was used to determine the risk factors of suicide. Conditional logistic regression is a specialized type of logistic regression used when case subjects with a particular condition are each matched with n control subjects without the condition and has become a standard for matched case-control data. In this study, we adopted a 1:1 matched case-control study design matching on age, gender and residence. In this conditioned multivariable logistic regression, the dependent variable was suicide (case $=1$, control $=0$), and the independent variable was impulsivity, while controlling for the following covariates: marital status, being left behind, social support, education, family annual income, and total stimulation of negative life. Physical diseases were not included in the regression because the total stimulation of negative life had included the effect of physical diseases. The adjusted OR and a 95% CI were used to assess the association between the risk factors and suicide. Finally, $p < 0.05$ was considered statistically significant.

## RESULTS

### Comparison of socio-demographic and psycho-social characteristics between suicides and living controls

Table 1 shows a comparison of socio-demographic and psycho-social characteristics between 242 pair of matched suicides and living controls. Compared to the control group, the suicide group were more likely to be not currently married (49.6% vs. 29.8%, $p < 0.001$), unemployed (80.6% vs. 69.8%, $p = 0.021$), being left behind (16.9% vs. 10.3%, $p = 0.034$),

**Table 1 Summary information of suicides and living controls.**

| Variables | Suicides | Living controls | $\chi^2$/F/z | $p$ |
|---|---|---|---|---|
| **N** | 242 | 242 | – | – |
| **Gender** | | | – | – |
| Male | 135(55.8) | 135(55.8) | | |
| Female | 107(44.2) | 107(44.2) | | |
| **Age (years)** | | | 1.12 | 0.569 |
| 60–69 | 73(30.2) | 70(28.9) | | |
| 70–79 | 100(41.3) | 111(45.9) | | |
| ≥80 | 69(28.5) | 61(25.2) | | |
| **Education** | | | 1.92 | 0.383 |
| Below primary school | 111(45.9) | 96(39.7) | | |
| Primary school | 105(43.4) | 116(47.9) | | |
| Above primary school | 26(10.7) | 30(12.4) | | |
| **Marital status** | | | 19.89 | <0.001 |
| Currently married[a] | 122(50.4) | 170(70.2) | | |
| Not currently married[b] | 120(49.6) | 72(29.8) | | |
| **Employment** | | | 7.84 | 0.020 |
| Employed | 40(16.5) | 59(24.4) | | |
| Unemployed | 195(80.6) | 169(69.8) | | |
| Retired | 7(2.9) | 14(5.8) | | |
| **Family annual income (CNY[c])** | | | 1.87 | 0.394 |
| <3600 | 88(36.4) | 74(30.6) | | |
| 3600–9999 | 88(36.4) | 98(40.5) | | |
| ≥10000 | 66(27.3) | 70(28.9) | | |
| **Being left-behind** | | | 4.49 | 0.034 |
| Yes | 41(16.9) | 25(10.3) | | |
| No | 201(83.1) | 217(89.7) | | |
| **Physical diseases** | | | 18.52 | <0.001 |
| Yes | 202(83.5) | 161(66.5) | | |
| No | 40(16.5) | 81(33.5) | | |
| **Total scores of negative life events** | 49.76 ± 30.19 | 25.24 ± 28.46 | **79.95** | **<0.001** |
| **Social support ( mean ±SD )** | 22.88 ± 5.98 | 27.47 ± 6.81 | **78.20** | **<0.001** |
| **BIS score[d]** | 98.79 ± 16.63 | 86.91 ± 15.18 | **78.99** | **<0.001** |
| Nonplanning impulsivity | 35.18 ± 7.24 | 31.07 ± 7.07 | **45.34** | **<0.001** |
| Motor impulsivity | 28.43 ± 8.28 | 23.70 ± 6.05 | **50.91** | **<0.001** |
| Attentional impulsivity | 35.18 ± 5.95 | 32.13 ± 5.67 | **41.35** | **<0.001** |

Notes.
[a] Included married and living with spouse or cohabiting.
[b] Included single, divorced, widowed and married but living apart.
[c] CNY, Chinese Yuan.
[d] BIS: Barratt Impulsiveness Scale.

and having physical diseases (83.5% vs. 66.5%, $p < 0.001$). The suicide group had higher score in negative life events (49.76 ± 30.19 vs 25.24 ± 28.46, $p < 0.001$) and lower score in social support (22.88 ± 5.98 vs 27.47 ± 6.81, $p < 0.001$) than living controls. The suicide

**Table 2** Comparisons of suicide behaviors by high and low impulsivity group.

| Variables | High impulsivity[a] | Low impulsivity | $\chi^2$/t/z | $p$ |
|---|---|---|---|---|
| **Suicides** | 120 (49.6) | 122 (50.4) | – | – |
| **Suicide time** | | | 0.71 | 0.447 |
| 6:00–17:59 | 95 (79.2) | 91 (74.6) | | |
| 18:00–5:59 | 25 (20.8) | 31 (25.4) | | |
| **Suicide place** | | | 0.19 | 0.700 |
| Home | 104 (86.7) | 108 (88.5) | | |
| Others | 16 (13.3) | 14 (11.5) | | |
| **Suicide means** | | | 1.199 | 0.945 |
| Pesticides | 63 (52.5) | 62 (50.8) | | |
| Hanging | 45 (37.5) | 50 (41.0) | | |
| Drowning | 5 (4.2) | 4 (3.3) | | |
| Non-pesticides poisoning | 5 (4.2) | 3(2.5) | | |
| Jumping off a building | 1(0.8) | 2(1.6) | | |
| Cutting throat | 1(0.8) | 1(0.8) | | |
| **History of attempted suicide** | | | **8.24** | **0.005** |
| Yes | 31(25.8) | 14(11.5) | | |
| No | 89(74.2) | 108(88.5) | | |
| **Suicide intent** | 4.93 ± 2.58 | 4.85 ± 2.41 | −0.23 | 0.821 |

Notes.
[a] High impulsivity was defined by BIS score above the median.

group also showed higher scores in overall BIS scores ($98.79 \pm 16.63$ vs. $86.91 \pm 15.18$, $p < 0.001$), as well as its three sub-scales (all $p$ values $< 0.001$).

## Comparison of suicidal behaviors between high and low impulsive suicides

Suicide cases were dichotomized into two groups, higher impulsivity and lower impulsivity group, based on the median of BIS scores. Characteristics of suicide behavior in these two groups were compared (Table 2). The proportion of previous suicide attempt was significantly higher in the higher impulsivity group than that in the lower impulsivity group (25.8% vs. 11.5%, $p = 0.005$), while no significant differences were found in time, location, or means of suicide, or suicide intent between the high and low impulsivity group.

## Multivariate logistic regression on risk factors of suicide

Multivariate logistic regression was used to determine the risk factor of suicide (Table 3). Independent variables included in the regression were marital status, being left-behind, income, years of school education, impulsivity, social support, and life events. Four variables entered the final model: impulsivity (OR: 1.03, 95% CI [1.01-1.06]), marital status (OR: 2.26, 95% CI [1.04–4.92]), social support (OR: 0.93, 95% CI [0.88–0.99]), and life events (OR: 1.02, 95% CI [1.01–1.03]). We also run additional regressions with each subscales of impulsivity as independent variable and found only motor impulsivity was independently associated with suicide in the final model (OR:1.08, 95% CI [1.03–1.13]).

**Table 3  Risk factors of suicide among rural elderly in China.**

| Independent variables | OR (95% CI) | $p$ |
| --- | --- | --- |
| **Impulsivity** | **1.03 (1.01–1.06)** | **0.015** |
| **Marital status** | | |
| Currently married | 1(ref) | |
| Not Currently married | **2.26 (1.04–4.92)** | **0.039** |
| **Being left-behind** | | |
| No | 1(ref) | |
| Yes | 0.63 (0.22–1.83) | 0.392 |
| **Social support** | **0.93 (0.88–0.99)** | **0.023** |
| **Education** | | |
| Below primary school | 1(ref) | |
| Primary school | 1.58 (0.43–5.86) | 0.491 |
| Above primary school | 0.72 (0.24–2.11) | 0.547 |
| **Family annual income (CNY[a])** | | |
| <3600 | 1(ref) | |
| 3600–9999 | 0.59 (0.24–1.43) | 0.241 |
| ≥10000 | 0.51 (0.21–1.25) | 0.143 |
| **Total scores of negative life events** | **1.02 (1.01–1.03)** | **0.001** |

**Notes.**
[a] CNY, Chinese Yuan.

# DISCUSSION

In the present study, we found that the suicide group had significantly higher scores in the total and three sub-scale scores of impulsivity than the control group. Within the suicide group, those with higher impulsivity were more likely to have a history of previous suicide attempt than those with lower impulsivity. Conditioned multivariable logistic regression showed that impulsivity was independently associated with suicide, after controlling for socio-demographic characteristics, social support, and life events.

Our major finding was that impulsivity was an independent risk factor for suicide among rural older adults in China. This finding was consistent with most studies conducted among youth in both urban and rural areas, and in China and abroad (*Florez et al., 2019*; *Swahn et al., 2012*; *Zhang et al., 2011*). *Zhang et al. (2011)* investigated suicide among rural youth from three provinces using case–control psychological autopsy method and found dysfunctional impulsivity as a risk factor for suicide. When compared with other studies conducted among rural elderly, our finding was in line with some studies while contrasted with others. *Neufeld & O'Rourke (2009)* examined impulsivity and hopelessness in predicting suicide-related ideation among older adults and found impulsivity played a stronger impact than hopelessness. However, in Liu et al.'s (*Liu, Qin & Jia, 2018*) research on comparing suicide risk factors between elderly suicides and non-elderly suicides, they found impulsivity as a risk factor for suicide in non-elderly, but not the elderly. While it has been widely recognized that impulsivity is a common feature of youth suicide, impulsivity in elderly suicide is more controversial and needs further research attention to understand

the underlying mechanism (*Demircin et al., 2011*). In general, our study finding indicates future development of suicide prevention programs among rural elderly need to pay attention to their impulsivity level, and provide timely assessment and effective treatment to reduce their risk of suicide.

This study compared suicidal behavior feature between higher impulsivity and lower impulsivity group. Impulsivity is associated with the history of attempted suicide, which is consistent with the finding of previous studies. Studies have showed that impulsive people are likely to attempt suicide repeatedly (*Mann et al., 1999*) and that people with repeated suicide attempts are more impulsive than those with first-time suicide attempts (*Dougherty et al., 2004*). A psychological autopsy study in youth suicides in China shows that the more impulsive population is prone to commit suicide through pesticide (*Zhang & Li, 2013*). This finding suggests past history of suicide as a risk factor for impulsivity, which may increase future risk of suicide. In future suicide prevention programs, more caution needs to be paid to individuals who had a past history of suicide attempt, with necessary screening of impulsivity and suicide risk conducted to these individuals and more psycho-social support provided to them. However, we do not find significant difference between the suicide means used by suicides with high and low impulsivity, with pesticides suicide being the most commonly means for both groups. The patterns of suicide methods in our study were in keeping with that in Liu et al.'s study on both elderly and non-elderly suicide (*Liu, Qin & Jia, 2018*). This finding suggests that easy access to pesticides in rural areas may be a potential facilitator for elderly suicide, regardless their impulsivity level. One implication may be that controlling suicide means such as limiting the purchase of lethal pesticides may serve as an effective preventive strategy for suicide.

Aside from impulsivity, our study also demonstrated other important risk factors related to suicide, such as low social support, not being currently married status, and experiencing more negative life events. This finding was in accordance with most of the previous studies showing poorer living conditions of the rural elderly such as living alone, lower social support and negative life events may aggravate their risk of suicide (*Liu, Qin & Jia, 2018*). The elderly, especially those who are not married, usually have few social network and receive little social support, which makes them more vulnerable to negative life events and more easily to take extreme risk behaviors such as suicide. It turns out that an abnormal psychological status and the instability of emotion and actions are responsible for the tendency to extreme behavior including suicide. All these findings indicate that suicide is not only the consequence of impulse ridden personality, but also be affected by complicated interaction of various risk factors (*Paris, 2005*). Implications of this finding include more social support, more psycho-education need to be provided to rural elderly to improve their psychosocial well-being, especially among those who had experienced significant negative life events.

In interpreting results of the present study, two major limitations should be noted. First, psychological autopsy is a retrospective method that relies on interviews with third-party informants, which has certain shortcomings such as sampling bias, recall bias, external confounders, and lack of standard procedure, and thus may lead to inaccurate assessment and unreliable conclusions (*Pouliot & De Leo, 2006*). However, psychological autopsy has

been in use for over 30 years as a prime approach in identifying risk factors for suicide and is the best available method so far. Besides, we used a case-control study design to collect the data from both the case and control groups using the same method, which may minimize the risks of external confounders. Future research may benefit from combining psychological autopsy with qualitative approaches and interviewing as many proxies of the suicides as possible to further strengthen its reliability and validity (*Hjelmeland et al., 2012*). Second, in examining risk factors of suicide, we didn't include some other well-known factors such as mental illness and prior suicide attempt. Although mental illness, especially mood disorder, has been identified as a most prominent predictor for suicide in western world, studies in Asian countries such as China showed different patterns where suicides without a diagnosis of mental illness were more common (*Milner, Sveticic & De Leo, 2013*). Future research that includes assessment on mental illness diagnosis in Chinese rural elderly suicide population and comparison with other countries is warranted to test such a difference. For previous suicide attempt, we found only a few cases of previous suicide attempt in living controls, which may result in low power of hypothesis testing if included in the multivariate analysis. Future research with larger sample size is needed to examine the association between precious suicide attempt and suicide death with more power.

## CONCLUSIONS

In conclusion, our finding showed that impulsivity was a significant independent risk factor of suicide among rural elderly in China, which warrants future research to further test such association and examine its underlying mechanism. Future suicide intervention programs may consider adding impulsivity assessment into its routine risk evaluation and treat high risk population in time and properly. In addition, social support may be provided to the elderly, especially those who experienced significant negative life events, to improve their psycho-social well-being and prevent suicide.

### Funding
The authors received no funding for this work.

### Competing Interests
The authors declare there are no competing interests.

### Author Contributions
- Yunfang Zhou performed the experiments, analyzed the data, prepared figures and/or tables, and approved the final draft.
- Zhenyu Ma and Cun-Xian Jia performed the experiments, authored or reviewed drafts of the paper, and approved the final draft.
- Liang Zhou conceived and designed the experiments, authored or reviewed drafts of the paper, and approved the final draft.

## Human Ethics

The following information was supplied relating to ethical approvals (i.e., approving body and any reference numbers):

Xiangya School of Public Health, Central South University granted Ethical approval to carry out the study within its facilities (Ethical Application Ref: CTXY-130041-1).

## Data Availability

Raw data are available as a Supplemental File.

## Supplemental Information

Supplemental information for this article can be found online at http://dx.doi.org/10.7717/peerj.11801#supplemental-information.

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

# PeerJ

**Klonsky ED, May AM. 2015.** Impulsivity and suicide risk: review and clinical implications. 32, (8) p. 13. Gale Academic OneFile. Psychiatric Times. (accessed on 14 March 2021).

**Li M, Katikireddi SV. 2019.** Urban-rural inequalities in suicide among elderly people in China: a systematic review and meta-analysis. *International Journal for Equity in Health* **18**:2 DOI 10.1186/s12939-018-0881-2.

**Li X, Xiao Z, Xiao S. 2009.** Suicide among the elderly in mainland China. *Psychogeriatrics* **9**:62–66 DOI 10.1111/j.1479-8301.2009.00269.x.

**Liu BP, Qin P, Jia CX. 2018.** Behavior characteristics and risk factors for suicide among the elderly in rural China. *Journal of Nervous and Mental Disease* **206**:195–201 DOI 10.1097/NMD.0000000000000728.

**Lu CF, Jia CX, Xu AQ, Dai AY, Qin P. 2012.** Psychometric characteristics of Chinese version of Barratt Impulsiveness Scale-11 in suicides and living controls of rural China. *Omega* **66**:215–229.

**Ma Z, He Q, Nie G, Jia C, Zhou L. 2020.** Reliability and validity of short Beck Hopelessness Scale in psychological autopsy study among Chinese rural elderly. *International Psychogeriatrics* **32**:525–531 DOI 10.1017/S1041610219001315.

**Mann JJ, Waternaux C, Haas GL, Malone KM. 1999.** Toward a clinical model of suicidal behavior in psychiatric patients. *American Journal of Psychiatry* **156**:181–189.

**McHugh CM, Lee RSChun, Hermens DF, Corderoy A, Large M, Hickie IB. 2019.** Impulsivity in the self-harm and suicidal behavior of young people: A systematic review and meta-analysis. *Journal of Psychiatric Research* **116**:51–60 DOI 10.1016/j.jpsychires.2019.05.012.

**Milner A, Sveticic J, De Leo D. 2013.** Suicide in the absence of mental disorder? A review of psychological autopsy studies across countries. *International Journal of Social Psychiatry* **59**:545–554 DOI 10.1177/0020764012444259.

**Mo Q, Zhou L, He Q, Jia C, Ma Z. 2019.** Validating the life events scale for the elderly with proxy-based data: a case-control psychological autopsy study in rural China. *Geriatrics & Gerontology International* **19**:547–551 DOI 10.1111/ggi.13658.

**Moeller FG, Barratt ES, Dougherty DM, Schmitz JM, Swann AC. 2001.** Psychiatric aspects of impulsivity. *American Journal of Psychiatry* **158**:1783–1793 DOI 10.1176/appi.ajp.158.11.1783.

**Neufeld E, O'Rourke N. 2009.** Impulsivity and hopelessness as predictors of suicide-related ideation among older adults. *Canadian Journal of Psychiatry. Revue Canadienne de Psychiatrie* **54**:684–692.

**Pan YF, Ma ZY, Zhou L, Jia CX. 2020.** Psychometric characteristics of duke social support index among elderly suicide in rural China. *Omega* **82**:105–119 DOI 10.1177/0030222818805356.

**Paris J. 2005.** The development of impulsivity and suicidality in borderline personality disorder. *Development and Psychopathology* **17**:1091–1104.

**Patton JH, Stanford MS, Barratt ES. 1995.** Factor structure of the Barratt impulsiveness scale. *Journal of Clinical Psychology* **51**:768–774 DOI 10.1002/1097-4679(199511)51:6<768::AID-JCLP2270510607>3.0.CO;2-1.

**Pouliot L, De Leo D. 2006.** Critical issues in psychological autopsy studies. *Suicide Life Threat Behav* **36**:491–510 DOI 10.1521/suli.2006.36.5.491.

**Sha F, Yip PSF, Law YW. 2017.** Decomposing change in China's suicide rate, 1990-2010: ageing and urbanisation. *Injury Prevention Journal of the International Society for Child & Adolescent Injury Prevention* **23**:injuryprev2016-042006.

**Smith AR, Witte TK, Teale NE, King SL, Bender TW, Joiner TE. 2008.** Revisiting impulsivity in suicide: implications for civil liability of third parties. *Behavioral Sciences and the Law* **26**:779–797 DOI 10.1002/bsl.848.

**Swahn MH, Ali B, Bossarte RM, Van Dulmen M, Crosby A, Jones AC, Schinka KC. 2012.** Self-harm and suicide attempts among high-risk, urban youth in the U.S.: shared and unique risk and protective factors. *nternational Journal of Environmental Research and Public Health* **9**:178–191 DOI 10.3390/ijerph9010178.

**Turecki G, Brent DA, Gunnell D, O'Connor RC, Oquendo MA, Pirkis J, Stanley BH. 2019.** Suicide and suicide risk. *Nature Reviews Disease Primers* **5**:74 DOI 10.1038/s41572-019-0121-0.

**Wang CW, Chan CL, Yip PS. 2014.** Suicide rates in China from 2002 to 2011: an update. *Soc Psychiatry Psychiatr Epidemiol* **49**:929–941 DOI 10.1007/s00127-013-0789-5.

**WHO. 2019.** World Health Organization. (2019). Mental health. *Available at https://www.who.int/gho/mental_health/en/*.

**Zhang J, Jia CX. 2007.** Validating a short version of the Suicide Intent Scale in China. *Omega* **55**:255–265 DOI 10.2190/OM.55.4.a.

**Zhang J, Li Z. 2013.** Characteristics of Chinese rural young suicides by pesticides. *International Journal of Social Psychiatry* **59**:655–662 DOI 10.1177/0020764012450995.

**Zhang J, Li N, Tu XM, Xiao S, Jia C. 2011.** Risk factors for rural young suicide in China: a case-control study. *Journal of Affective Disorders* **129**:244–251 DOI 10.1016/j.jad.2010.09.008.

**Zhou L, Wang GJ, Jia CX, Ma ZY. 2019.** Being left-behind, mental disorder, and elderly suicide in rural China: a case-control psychological autopsy study. *Psychological Medicine* **49**:458–464 DOI 10.1017/S003329171800106X.