# Peer review of "Relationship between impulsivity and suicide among the rural elderly in China: a case-control psychological autopsy study"

_PeerJ, doi:10.7717/peerj.11801_

## Round 0.1 · original submission · Major Revisions

Thank you for submitting this interesting study. However, the manuscript will require major revision based on the reviewers’ comments, which is attached. Please kindly address these issues accordingly.

Reviewer 1 ·

Basic reporting

The manuscript discussed the association between impulsivity and suicide death in elderly population in China. The paper is clear and well written. Authors reviewed related literatures sufficiently.

Experimental design

It's an excellent psychological autopsy case-control study. The hypothesis was well defined.

Validity of the findings

It's acceptable. The conclusions were clear.

Additional comments

Mental illness and prior suicide attempt are well-known risk factors of suicide death in China and other countries. Could authors verify why the two important variables were not included in the multivariate analysis?

Reviewer 2 ·

Basic reporting

Although the paper is readable, I think it can be improved and written in a more concise way.
Line 63 TL. 2006 and line 70 Liu Z 2017. Please check the two references.
Line 76, "expression of suicide" is problematic.
Line 87, "negative" is inaccurate here, It could be "negatively associated with". But in the following example, it seems that the authors mean "no association".
Line 97-102 and line 106-107. The two paragraphs repeated the design of the study and objective as well.
Line 112, "died by suicides" is problematic here.
Line 138-139, the two terms, "marital stability" and "marital instability", seem problematic. I suggest the authors not to use the two to define two different marital status. The terms used in Tables are appropriate.
Line 138, please consider "the status of being left-behind".

Experimental design

See my general comments as below.

Validity of the findings

See my general comments as below.

Additional comments

1. Abstract. Please specify how case and control groups were matched and how impulsivity was assessed. Please also add the conclusion of this study here. Please consider whether it is appropriate to say "impulsivity increased the risk of suicide". This is a case-control study and data were collected cross-sectionally, I do not think this study design can answer the causality question.
2. Line 87-96, I do not think the study "Wyder & De Leo 2007" can indicate no relationship between attempt and impulsivity, because no comparison between subjects with and without attempt. Please review studies on impulsivity and completed suicide here, because non-fatal suicidal behaviors, including attempted suicides, are different from completed suicide.
3. Measurements. For deaths by suicides, their data were collected from informants or proxies. Please specify this and consider whether these instruments are also valid and reliable when scales were not completed by themselves per se.
4. Table 2, means of suicide. I think it is too crude to dichotomize it into pesticides vs. others. The comparison results might be statistically significant if more subtypes were compared.

Reviewer 3 ·

Basic reporting

Need modifications on some English expressions. Overall clear messages can be seen.

Experimental design

Research question well defined, but would be good if with better justifications on research gap and significance.

Validity of the findings

Yes. The method and also the findings seems novel. But would be good if can be compared with other populations, maybe also consider some other factors, and show clearly the uniqueness and importance of the findings.

Additional comments

Title: Relationship between impulsivity and suicide among the rural elderly in China: A case–control psychological autopsy study

This is a useful study using psychological autopsy among rural older adults in China. The method is sound, however, the findings and the rationale of the study, as well as the discussion need further modifications.

Abstract:
1. “The relationship between impulsivity and suicide is inconsistent in different populations. Hence, the relationship between impulsivity and suicide still needs to be studied among the elderly population.” ---This sentence looks less convincing as the rationale for the study?
2. There is no conclusion in abstract? Please check if this complies with Journal requirement.
3. A first impression is the OR is quite low 1.03 (95%CI: 1.01-10.6) though it is independent after controlling for the effects of marital status, education, family annual income, being left behind, social support, and negative life events. (Have other factors controlled as well, e.g. depression, anxiety?)
Background
4. “Suicide, which is the act of intentionally taking one’s own life (TL 2006), has been considered an international public health problem. The global prevalence of suicide has remarkably dropped in recent years. However, suicide still has resulted in 828,000 global deaths in 2015(Mortality & Causes of Death 2016).” ---Maybe a better flow is : “Suicide, which is the act of intentionally taking one’s own life (TL 2006), has been considered an international public health problem. Suicide still has resulted in 828,000 global deaths in 2015(Mortality & Causes of Death 2016), although its global prevalence has remarkably dropped in recent years.”
5. “Moreover, the suicide rates in rural areas are higher than those in urban areas in China (Liu Z 2017)” --- Is this referring to the older adults or general population in rural areas?
6. Would the authors explain more why it is important to look into the relationship of impulsivity and suicide, especially among the older adults? The rationale of this study, in a clearer way?

Methods
7. Please rewritten the sentence “Information on died by suicides among the elderly”---- “Information on death by suicides”?
8. May need to clarify who are “We” in this sentence “We received training on psychological autopsy and interview methods before the start of the interviews.”
9. It might be worthy mentioning that the interview for “the suicidal victims” group was conducted among the informants in methods, otherwise, a bit confusing, although a previous study covering the methods has been referred to.
10. Maybe the authors want to describe a bit more on their statistical method: Conditioned multivariable logistic regression ---How it has been applied in their study.
11. It may be better to also mention the results of the three subscales of BIS in text (since the authors have mentioned in the first paragraph). Were the three subscale scores also independently associated with suicide? (Were they significant in the final model)?

Discussion
12. “Impulsivity increases the risk of suicide among the rural Chinese elderly, which is consistent with previous studies (Fresan et al. 2016; Liu et al. 2018; Maser et al. 2015; Shelef et al. 2018).” –The authors may want to show this is consistent with some of the previous studies (as mentioned in the background).
13. Overall the 3rd paragraph in Background, and the 2nd paragraph in the Discussion section seem showing similar information but it could be better if the background could show better rationale, and the discussion could show a better deep look at the results and comparisons?
14. Limitations --- seems other important factors are not included and adjust in this study, e.g. mental disorders/depression. From the study by Zhou et al. in 2019, other factors are included and found to be significant (e.g. Independent risks of suicide included unstable marital status [odds ratio (OR) 4.19, 95% confidence interval (CI) 1.61–10.92], unemployed (compared with employed, OR 4.43, 95% CI 1.09–17.95), depressive symptoms (OR 1.34, 95% CI 1.21–1.48), and mental disorder (OR 6.28, 95% CI 1.75–22.54). Structural equation model indicated that the association between being left-behind and suicide was mediated by mental disorder, depressive symptoms, stressful life events, and social support.) Are there specific reasons?
15. How would the results / significance of impulsivity among the rural older adults compared to other populations, e.g. younger / adult population, and urban areas?
16. The impulsivity was not related to suicide time, place and means. Was this consistent with or different from previous findings? The authors mentioned that it was different from the youth? Are there any other reasons beyond age?
17. “Our results indicate that suicides with different level of impulsivity may need different preventive strategy.” ---Can this be explained more?
18. “In future research on suicide prevention, the family and society should pay increasing
attention to impulsivity and learn to handle them differently. People with impulsivity traits should be encouraged to consider and positively face life’s difficulties to effectively reduce or prevent the occurrence of suicide.” ---Can the authors explain more how the findings imply future services and practices?

---

## Round 0.2 · accepted · Accept

I agree with the two reviewers.

Reviewer 1 ·

Basic reporting

The revised manuscript is well written. Authors addressed reviewers' concerns well.

Experimental design

Good psychological autopsy case-control study.

Validity of the findings

Acceptable.

Additional comments

All my concerns has been well addressed.

Reviewer 2 ·

Basic reporting

No further comments

Experimental design

No further comments

Validity of the findings

No further comments

Additional comments

No further comments